# Characteristic Analysis of Featured Genes Associated with Cholangiocarcinoma Progression

**DOI:** 10.3390/biomedicines11030847

**Published:** 2023-03-10

**Authors:** Qigu Yao, Wenyi Chen, Feiqiong Gao, Yuchen Wu, Lingling Zhou, Haoying Xu, Jong Yu, Xinli Zhu, Lan Wang, Lanjuan Li, Hongcui Cao

**Affiliations:** 1State Key Laboratory for Diagnosis and Treatment of Infectious Diseases, Collaborative Innovation Center for Diagnosis and Treatment of Infectious Diseases, The First Affiliated Hospital, Zhejiang University School of Medicine, 79 Qingchun Road, Hangzhou 310003, China; yaoqigu@zju.edu.cn (Q.Y.);; 2Department of Radiation Oncology, The First Affiliated Hospital, Zhejiang University School of Medicine, 79 Qingchun Road, Hangzhou 310003, China; 3Key Laboratory of Diagnosis and Treatment of Aging and Physic-Chemical Injury Diseases of Zhejiang Province, 79 Qingchun Road, Hangzhou 310003, China; 4Jinan Microecological Biomedicine Shandong Laboratory, Jinan 250117, China

**Keywords:** cholangiocarcinoma, hub gene, tumorigenic mechanism, prognostic biomarkers, weighted gene co-expression network analysis

## Abstract

The noninvasive diagnosis of cholangiocarcinoma (CCA) is insufficiently accurate. Therefore, the discovery of new prognostic markers is vital for the understanding of the CCA mechanism and related treatment. The information on CCA patients in The Cancer Genome Atlas database was used for weighted gene co-expression network analysis. Gene Ontology (GO) analysis and Kyoto Encyclopedia of Genes and Genomes (KEGG) pathway analysis were applied to analyze the modules of interest. By using receiver operating characteristic (ROC) analysis to analyze the Human Protein Atlas (HPA), the featured genes were subsequently verified. In addition, clinical samples and GSE119336 cohort data were also collected for the validation of these hub genes. Using WGCNA, we identified 61 hub genes that regulated the progression and prognosis of CCA. Eight hub genes (VSNL1, TH, PCP4, IGDCC3, RAD51AP2, MUC2, BUB1, and BUB1B) were identified which exhibited significant interactions with the tumorigenic mechanism and prognosis of CCA. In addition, GO and KEGG clarified that the blue and magenta modules were involved with chromosome segregation, mitotic and oocyte meiosis, the cell cycle, and sister chromatid segregation. Four hub genes (VSNL1, PCP4, BUB1, and BUB1B) were also verified as featured genes of progression and prognosis by the GSE119336 cohort data and five human tissue samples.

## 1. Introduction

Cholangiocarcinomas (CCAs) are epithelial tumors that arise from the intrahepatic bile duct or large bile ducts [1]. In addition, CCA is very common in digestive system tumors, and its incidence is second only to hepatocellular carcinoma [2]. Over the past 40 years, the global incidence rate of CCA has increased to approximately 18% [3]. Early diagnosis of CCA remains challenging because most patients have no conspicuous symptoms during the early stages of disease [4]. Furthermore, highly desmoplastic, paucicellular tumors develop in the liver or large bile ducts, thereby limiting the sensitivity of pathological diagnosis [5]. Given the intertumoral and intratumoral heterogeneity of CCA, key therapeutic targets have not been defined for this disease [5,6]. Therefore, reliable biomarkers are needed for the early diagnosis of CCA, and important prognostic factors should be identified.

With the increasing popularity of next-generation sequencing technology, targeted or operable molecular changes in transcription levels can be identified [7]. Many cancers have been analyzed using this technique to detect novel biomarkers for tumor diagnosis, as well as to mitigate the recurrence and inhibition of drug resistance [8,9]. Based on public database findings or sequencing data from clinical samples, Li et al. elucidated nine hub genes highly related with the pathological T stage of hepatocellular carcinoma [10]. Similarly, utilizing weighted gene co-expression network analysis (WGCNA) to construct an endogenous competitive RNA regulatory network, Peng et al. identified six possible pivotal genes in metformin in the treatment of diabetes and colorectal cancer [11].

Common CCA screening indicators, such as carbohydrate antigen 199, cancer antigen 125, and cancer antigen −50, are not sensitive or specific enough [12]. For example, CA199 is also highly expressed in pancreatic cancer, hepatobiliary carcinoma, gastric cancer, and colorectal cancer [13,14]. Additionally, the levels of CA125 increase in patients with fallopian tube adenocarcinoma, ovarian cancer, pancreatic cancer, colorectal cancer, and breast cancer [15,16,17]. There are obvious genomic changes in the pathogenesis of CCA, and the study of CCA may be helpful in finding their potential diagnostic and prognostic value. For example, MRPS18A, CST1, and SCP2 were associated with the pathological stage, liver function, and overall survival in CCA patients [18]. The inhibition of HER2/neu was identified as a promising treatment target for patients with CCA [19]. In approximately 60% of intrahepatic CCAs, the FGFR2 and the genes encoding isocitrate dehydrogenases might be suitable therapeutic targets [20]. Nevertheless, further research is needed concerning novel biomarkers for clinical decision making.

Here, patients with CCA in The Cancer Genome Atlas (TCGA) were used to perform WGCNA. WGCNA was applied to find the correlation between the clinical information and the genes in the microarray samples [21]. Additionally, correlation networks were used to analyze the candidate biomarkers with Gene Ontology (GO), Kyoto Encyclopedia of Genes and Genomes (KEGG) analysis, and protein–protein interaction (PPI) networks. Overall, the identification of the modules and featured genes in CCA will help to build a precise approach to diagnosis and treatment.

## 2. Materials and Methods

### 2.1. Processing of Gene Expression Data

RNA sequencing data were collected by the Illumina HiSeq 2000 RNA Sequencing platform and then mean-normalized (per gene) across all TCGA cohorts. Phenotype information was also downloaded from UCSC Xena, accessed on 17 January 2021 (https://xenabrowser.net). The following inclusion criteria were used: (1) samples with RNA sequencing data available and (2) matched samples with adequate clinical data. Gene matrix profiles were cleaned and normalized using NetworkAnalyst, accessed on 22 February 2021 (https://www.networkanalyst.ca). A |log_2_fold change| > 1.0 and an adjusted *p*-value < 0.05 were set as the threshold for differentially expressed genes (DEGs). Ultimately, we included 36 bile duct cancer tissues and 9 matched normal tissues. Clinical characteristics were extracted, including age, albumin value, CA199 expression, cancer history in first-degree relatives, Child–Pugh grade, creatinine value, fetoprotein value, fibrosis score, neoadjuvant treatment history, neoplasm histological grade, pathological M, N, and T scores, perineural invasion, neoplasm cancer, platelet count, prothrombin time, family cancer history, vascular tumor cell type, survival, sex, ethnicity, vital status, prior malignancy, prior treatment, tumor stage, body mass index (BMI), height, and weight.

### 2.2. Gene Co-Expression Network Construction

In order to construct a co-expression network, the “WGCNA” package statistical software, accessed on 26 February 2021. (version 4.0.2, https://www.r-project.org) was employed to calculate the similarities between the gene expression profiles [21]. Outlier samples were identified by the averaging method in the “hclust” function within the “WGCNA” package. These outliers were then removed. Scale-free networks were constructed using the WGCNA method. The scale-free network is defined as a network in which most nodes in the network connect with a few other nodes, but a few nodes connect with many nodes. These networks are more tolerant of unexpected faults compared with networks in which most nodes connect with many other nodes. Using hierarchical clustering, the adjacency matrix was changed into a topological overlap matrix (TOM); the dynamic tree cut method was then used to identify different modules. When setting the parameters, the minimum number of genes contained in the network module was set at 30 (min module size = 30); the threshold value of the gene reclassification between modules was set at 0 (reassign threshold = 0); and the degree of dissimilarity was set at 0.25 (merge cut height = 0.25).

### 2.3. Construction of CCA Modules with Clinical Relationships

Pearson correlation coefficients between the module eigengenes (MEs) and the clinical information were calculated. Following the calculation of the core modules, gene significance (GS) and module membership (MM) were used to measure the association levels of the genes with clinical traits and the MEs. Module core genes were considered to have MM > 0.8 and GS > 0.2.

Combined with the hub genes identified by MM and GS, a q-weighted cutoff < 0.01 was settled to find featured genes using the network screening function. The common hub genes were determined by two methods and then visualized using Venn diagrams.

### 2.4. Functional Annotations for DEGs

The clusterProfiler package (version: 3.16.1) was adopted for GO biological function and KEGG signaling pathway analysis for the DEGs. The GO function and KEGG pathway used the default parameters in the clusterProfiler package, and the cutoff threshold was set to *p* < 0.0001.

### 2.5. Hub Gene Validation

The R package “survival” and Gene Expression Profiling Interactive Analysis (http://gepia.cancer-pku.cn), accessed on 27 February 2021, were employed to identify the key prognostic genes by means of Kaplan–Meier analysis. The Human Protein Atlas database (https://www.proteinatlas.org), accessed on 22 February 2021, was employed to validate the immunohistochemistry results of the featured prognostic genes [22].

### 2.6. Identification of Hub Gene Functional Annotations

Using the GO function and KEGG pathway enrichment analysis, we mapped, integrated, and visualized the characteristic genes and related tumor progression mechanisms. The GOplot R package (version: 1.0.2) was employed for visualization.

### 2.7. Human Samples

All the patients with CCA gave their informed consent, and the experimental design met the ethical requirements, which were approved by the Ethics Committee of The First Affiliated Hospital of Zhejiang University (No. 2014-272). Tissues, including para-tumor and CCA samples, were sampled immediately in the First Affiliated Hospital, Zhejiang University School of Medicine. 

### 2.8. Immunohistochemical Staining

Using xylene and alcohol, 5 µm thick human sample tissues were deparaffinized. The samples were then repaired with 3% H_2_O_2_-CH_3_OH for 10 min. Subsequently, they were submerged in a pH 9.0 buffer for antigen retrieval and then incubated at 37 °C overnight with primary antibody, BUB1B (Abcam, ab183496, 1:100), BUB1 (Abcam, ab195268, 1:100), PCP4 (Proteintech, 14705-1-AP, 1:200), or VSNL1 (Proteintech, 67134-1-IG, 1:200). After the sections had been washed with phosphate-buffered saline three times, they were submerged with horseradish peroxidase-conjugated IgG, rabbit anti-mouse IgG (Abcam, ab6728), and goat anti-rabbit IgG (Abcam, ab6721) at 37 °C for 40 min. Finally, the sample sections were developed with a 3,3′-diaminobenzidine tetrahydrochloride detection system kit (Abcam, ab64238) and hematoxylin staining solution (Servicebio, G1004). Protein expressions in the sections were detected via and analyzed by NDP.view2 (version 2.6.8). Five visual fields were randomly selected to compare the protein expression levels between the CCA and para-tumor samples. Quantitative analyses were conducted by Image J software. Protein expressions that were specifically located within cholangiocarcinoma cells or bile duct cells were compared. Notably, staining areas within normal hepatocytes were excluded. Mean optical density was chosen as a comparative indicator and could be calculated as follows: Mean optical density = Integrated option density/Area.

### 2.9. Construction of PPI Network

Using the Search Tool for the Retrieval of Interacting Genes, we evaluated the correlation and biological function of related genes [23]. The cutoff criterion was regarded as a confidence score ≥ 0.4. Cytoscape is a visualization software platform (version: 3.7.1; https://cytoscape.org), accessed on 22 February 2021, which can help to visualize the PPI network [24].

### 2.10. Statistical Analysis

R (version 4.0.2) was used for data analysis. Wilcoxon and Kruskal–Wallis tests were used for gene difference analysis. In this study, overall survival analysis was evaluated by the Kaplan–Meier method and log-rank testing. In the absence of special labels, *p* < 0.05 was regarded as significant in all statistical analyses.

## 3. Results

### 3.1. Data Collection and Processing

The overall flowchart of the study protocol is shown in Figure 1. The selected clinical information and mRNA were downloaded from UCSC Xena, including 36 bile duct cancer tissues and nine matched normal tissues (Appendix A). After data cleaning and normalization (removal of samples with repetition or missing information), 544 up-regulated genes and 1069 down-regulated genes met the screening conditions (Figure 2A,B). t-Distributed stochastic neighbor embedding of the DEGs demonstrated that the tumor and nontumor tissues were clearly distinguished (Figure 2C). The threshold of the adjusted *p*-value was < 0.05 and the |log_2_fold change was | > 1.0 for WGCNA and the subsequent analyses (Figure 2D). The corresponding clinical information is shown in Appendix A.

For WGCNA, the ‘hclust’ function clustered 45 samples and removed the outliers. As shown in Figure 2E, the parameters “cutHigh” and “minSieze” were set at 55 and 10, which ultimately yielded 28 samples.

### 3.2. Functional Annotations of Hub Genes

The clusterProfiler and enRichment packages were used to examine the biological function of the selected modules by GO analysis and KEGG pathway analysis (Appendix A). In the GO analysis, the top five GO terms were monooxygenase, passive transmembrane transporter, metal ion transmembrane transporter, channel activity, and ion channel (Appendix A). In KEGG analysis, the top five biological pathways were neuroactive ligand–receptor interaction, chemical carcinogenesis, metabolism of xenobiotics by cytochrome P450, drug metabolism–cytochrome P450, and steroid hormone biosynthesis (Appendix A).

### 3.3. Weighting Coefficient β Selection

The most important co-expression networks are scale-free networks, such that P (k)∼k − 1 (Appendix A) [25]. As the soft threshold (β) increases, the correlation coefficient of the network increases (R^2^ > 0.8) (Figure 3A), and the mean connectivity of the network subsequently decreases (Figure 3B). Therefore, β = 5 was selected to construct the co-expression network, such that log(k) was correlated with log[P(k)] (R^2^ = 0.87, slope = −1.33) (Figure 3C,D).

### 3.4. Co-Expression Network Construction

Adjacencies among the DEGs were evaluated, and then, the results were plotted in a TOM-based hierarchical gene clustering tree (Appendix A). Initially, the MEs were determined by principal component analysis. The Dynamic Tree Cut package is a function used to analyze modules of interest; the parament of the minimum module size was defined as 30 and the deepSplit was defined as 2 (Appendix A). Using the Dynamic Tree Cut Merged Dynamics package, the gene topological matrix was modified (Figure 4A). To reduce the complexity of the network, the MEs with a similarity > 0.75 were merged, and the numbers of modules were not changed. The eigengene dendrogram and heatmap were evaluated using the groups of correlated eigengenes and the dendrogram (Figure 4B). 

### 3.5. Identification of Clinical Modules

As shown in Figure 4C, the clinical modules were clustered, and their correlations were analyzed with a sample dendrogram. With respect to the clinical data, red represents a high value, white means a low value, and gray means a missing value. Through the R value and *p*-value, the connection between the MEs and the clinical characteristics was continuously scored, and the 13 modules were related to the clinical characteristics given above, as shown in Figure 4D. The blue and magenta modules were significantly associated with tumor stage (R = 0.38, *p* = 0.05; R = 0.47, *p* = 0.01). The whole genes of the blue and magenta modules are shown in Appendix A.

### 3.6. Identification of Hub Genes

The GS for the tumor stage and MM values for the genes in each module was calculated in order to identify the MM values associated with tumor differentiation, as shown in Figure 5A. The magenta module (correlation coefficient = 0.6, *p*-value = 5.1 × 10^−9^) and the blue module (correlation coefficient = 0.49, *p*-value = 3.6 × 10^−11^) were consistent with the adverse clinical characteristics of the CCA patients (Figure 5B,C). To identify the hub genes for tumor differentiation, a GS > 0.2 for the tumor stage and an MM > 0.8 were concurrently satisfied in two modules. This ruled out a total of 73 hub genes. Furthermore, the hub genes were required to meet the screening criteria (q-weighted < 0.01) of the ‘networkScreening’ function. In total, 61 featured genes were identified, as shown in the Venn plot (Figure 5E).

### 3.7. Functional Enrichment Analyses of the Hub Genes in Hub Modules

Applying GOplot packages, we investigated the biological significance of selected modules by using GO terms. In the whole genes of the blue and magenta modules, the top 10 biological processes in the GO analysis mainly focused on mitochondria (Figure 5D). Applying GOplot packages, we investigated the GO terms of the selected 61 hub genes, as well as the KEGG pathway analysis. The GO analysis results illustrated that the selected 61 hub genes were particularly enriched in chromosome segregation, mitotic sister chromatid segregation, and mitotic nuclear division (Figure 5F). According to the KEGG plot, the most important pathways were oocyte meiosis, cell cycle, dopaminergic synapse, alcoholism, and the Rap1 signaling pathway (Figure 5G). A co-expression network analysis was used to explore the degree of association among the 61 featured genes, with a coefficient of ≥0.53 (Appendix A). Next, we divided the two subgroups according to the correlation degree of the genes; the subgroups contained 23 and 33 genes, respectively (Appendix A). Among them, we found that the correlation coefficient between CALML3 and ALPP was the highest at 0.99 (Figure 6A). Based on the pathways enriched, we established connections between the hub genes and the cell cycles (Figure 6B and Appendix A).

### 3.8. Construction of PPI Network and Verification of Hub Genes

A PPI network was constructed using 61 featured genes, which had three subgroups. The denser and bigger the nodes are, the more important the gene is. Additionally, the thicker the edge, the closer the connection between the genes. The biggest subgroup had 28 nodes and 246 edges (Figure 6C). The remaining two subgroups had three nodes and four nodes, and two edges and three edges, respectively (Figure 6D).

### 3.9. Survival Analysis

Survival analysis is one of the most important indicators of prognosis. After analyzing the survival curve of 61 featured genes, 27 featured genes were related to the overall survival rate of the CCA patients (*p* < 0.05) (Appendix A).

### 3.10. Hub Gene Analysis and Validation

In order to further the potential mechanisms and roles of the 61 hub genes in CCA prognosis and tumor progression, receiver operating characteristic (ROC) analysis was employed. The results demonstrated that the area under the curve (AUC) of five of the hub genes in the 61 hub genes was >0.75, which could be used to distinguish CCA tumorigenic progression and prognosis (Figure 7A; Appendix A). In addition, five other hub genes (0.75 > AUC > 0.7) could be used to distinguish CCA tumorigenic progression (Appendix A). 

In Figure 7B, by combining 61 hub genes, the selected 10 hub genes (AUC > 0.7) in the ROC analysis for tumor progression, and 27 hub genes (*p* < 0.05) in the Kaplan–Meier survival curve, a Venn plot was constructed to find the genes which affected tumor development, as well as the prognosis and survival of the tumor patients. Eight hub genes were identified: VSNL1, TH, PCP4, IGDCC3, RAD51AP2, MUC2, BUB1, and BUB1B. These genes affected tumor development, as well as the prognosis and survival of the tumor patients (Figure 7B). In addition, VSNL1, RAD51AP2, PCP4, and MUC2 were correlated with tumorigenic progression by staging plot levels in the Gene Expression Profiling Interactive Analysis database (Appendix A).

We selected VSNL1, PCP4, BUB1, and BUB1B for verification at the mRNA and protein levels. Compared with nontumor tissues, the mRNA expression levels of PCP4, BUB1, and BUB1B were strikingly higher in the CCA patients. Additionally, the mRNA expression level of VSNL1 was strikingly reduced in the CCA tissue (Figure 7C). These results are similar to those of other GEO datasets (Figure 7D). We then used immunohistochemical staining to identify protein expression in the human samples. Then, PCP4 was verified by the HPA database. Furthermore, the experimental results of the immunohistochemistry showed that BUB1, BUB1B, and PCP4 were darker brown in the CCA tissue than in the normal tissue, while VSNL1 was the opposite. The mean optical density of the expression of BUB1, BUB1B, PCP4 and VSNIL1 in the CCA and adjacent tissue samples showed that the protein levels of BUB1, BUB1B, and PCP4 in the CCA tissue were significantly increased, but the protein level of VSNL1 in the CCA tissue was significantly decreased (Figure 8).

## 4. Discussion

CCA is a highly heterogeneous and rapidly developing disease, which is difficult to find early and hard to resection [26]. CCA heterogeneity causes poor treatment responses in most patients; so, surgery remains the primary treatment option [27]. However, with the development of new immunotherapy and biomarker therapy, the therapeutic effect on patients with end-stage CCA has also been improved [28]. Biomarkers can be used for accurate diagnosis, prediction of prognostic effects, and improvement of clinical efficacy [29]. 

Our study used personalized bioinformatics to identify the mRNA related to clinical features; our hope was to find new therapeutic targets or prognostic markers. Thus, we used the WGCNA method to identify 1613 DEGs from 45 CCA samples and constructed 13 co-expression modules. Furthermore, WGCNA was employed to analyze the relationships between the co-expression modules and the clinical characteristics by means of correlation coefficients.

The genes in clinically relevant modules are presumed to be functionally associated with each other, and this may be helpful for subsequent detection or treatment. In addition, according to the requirements of WGCNA, the number of multiple samples should be more than 15 [21]. Therefore, our WGCNA network is reliable. From 13 co-expression modules, the magenta (cor = 0.6) and blue (cor = 0.49) modules were extracted because of their associations with tumor-stage diagnosis. Through Venn plot analyses, we selected 61 genes that were highly correlated with CCA progression and prognosis. Subsequently, GO and KEGG analyses were used to functionally annotate these 61 hub genes. The GO analysis showed that two modules were particularly focused on chromosome segregation, mitotic sister chromatid segregation, and mitotic nuclear division. Condensed chromosome outer kinetochore and protein kinase activity were enriched in the cellular component and molecular function processes. The related pathways were mainly concentrated in the oocyte meiosis, cell cycle, dopaminergic synapse, alcoholism, and Rap1 signaling pathways. Loss of mitotic regulation is an important feature of cell carcinogenesis which leads to abnormal mitosis and chromosome segregation. [30]. Our results indicated that mitosis, meiosis, and the cell cycle are key factors that influence the speed of tumor deterioration, the same as those of Du et al. [31].

Furthermore, we identified eight hub genes (VSNL1, TH, PCP4, IGDCC3, RAD51AP2, MUC2, BUB1, and BUB1B) that promoted tumor progression and were associated with prognosis according to the Kaplan–Meier and ROC curve analyses. TH, IGDCC3, RAD51AP2, and MUC2 are genes that were identified by WGCNA and are related to the clinical prognosis and survival of cholangiocarcinoma, but they have already been reported [5,32,33,34]. Therefore, VSNL1, PCP4, BUB1, and BUB1B were chosen for further verification.

VSNL1 is related to neuronal calcium sensor proteins which are highly expressed in the cerebellum [35]. The research concerning VSNL1 has mainly focused on neurological diseases such as Alzheimer’s disease and medulloblastoma [36,37]. VSNL1 has also been investigated in gastrointestinal tumors, where it promotes the rapid growth and metastasis of gastric cancer cells [38], lymph node metastasis, and the deterioration of colorectal cancer [39]. However, the VSNL1 function in CCA needs to be further analyzed. Considering that some studies have shown that the high expression of VSNL1 in colorectal cancer is related to the high degree of tissue differentiation, it is speculated that the function of VSNL1 may be related to the metastasis of cholangiocarcinoma.

PCP4 encodes a modulator of calmodulin-mediated calcium binding, mainly expressed in brain, kidney, colon, and thyroid tissue. The main function of PCP4 is believed to be related to the regulation of the calmodulin-mediated signal [40]. The role of PCP4 is under investigation in neurological and endocrine diseases, such as polycystic ovary syndrome and Alzheimer’s disease. In adrenal adenomas, the overexpression of Pcp4 may be related to DNA methylation and may affect aldosterone secretion [41]. PCP4 is rarely studied in gastrointestinal tumors. PCP4 participates in the calmodulin-dependent kinase signaling pathway to inhibit cell apoptosis, and CCA may promote cancer cell adhesion, migration, and invasion.

BUB1 and BUB1B encode the serine/threonine–protein kinases that greatly contribute to mitosis [42,43]. At present, it is believed that the function of BUB1 is to activate the related genes of the spindle checkpoint by phosphorylating the related members of the mitotic checkpoint complex so as to affect the process of cell proliferation [44]. BUB1B is a checkpoint which is similar to BUB1. Additionally, BUB1B triggers polyploid cell apoptosis and inhibits the activity of PLK1 in the mitotic interphase [45]. Overall, it participates in two crucial aspects of mitosis: the spindle assembly checkpoint signal and the cell chromosome arrangement in mitosis. BUB1 is reportedly overexpressed in pancreatic ductal adenocarcinoma, gastric cancer, and multiple myeloma [46,47,48]. BUB1B has been studied in multiple tumors, such as colorectal carcinoma, CCA, and breast cancer [49,50,51]. Qiu reported that BUB1B may promote the proliferation of liver cancer cells by raising the mTORC1 signaling pathway [52]. BUB1 and BUB1B mediate apoptosis in response to chromosomal aberration and inhibit the proliferation and metastasis of tumor cells. BUB1 and BUB1B may accelerate the process of cholangiocarcinoma by promoting the mitosis of cholangiocytes.

This study also has some shortcomings. First of all, despite the use of clinical samples and TCGA samples, the number of samples used was still small in general, and the sample size should be expanded in a follow-up study. Second, the CCA hub genes and their relationships with tumor progression and prognosis require experimental verification, both in vitro and in vivo. Third, CCAs are epithelial tumors which can be divided into three subtypes. Fourth, scoring these genes in the CCA library using the data in the single cell map or single cell portal will help to explore the changes in these hub genes in the tumor microenvironment. Therefore, our study results might be more convincing if the samples were divided into several subgroups. Therefore, additional clinical samples with detailed clinical data are needed to confirm our findings. Basic research concerning hub genes is important for accurate identification of the mechanisms underlying tumor progression and prognosis.

## 5. Conclusions

Through WGCNA, GO, and KEGG analyses, ROC analysis, and immunohistochemical staining, we identified eight hub genes (VSNL1, TH, PCP4, IGDCC3, RAD51AP2, MUC2, BUB1, and BUB1B) that may contribute to CCA progression and prognosis. Of these eight genes, VSNL1, PCP4, BUB1, and BUB1B may also serve as novel biomarkers for the determination of disease progression. Additional mechanistic research and clinical verification may contribute to individualized treatment in the future.

## Figures and Tables

**Figure 1 biomedicines-11-00847-f001:**
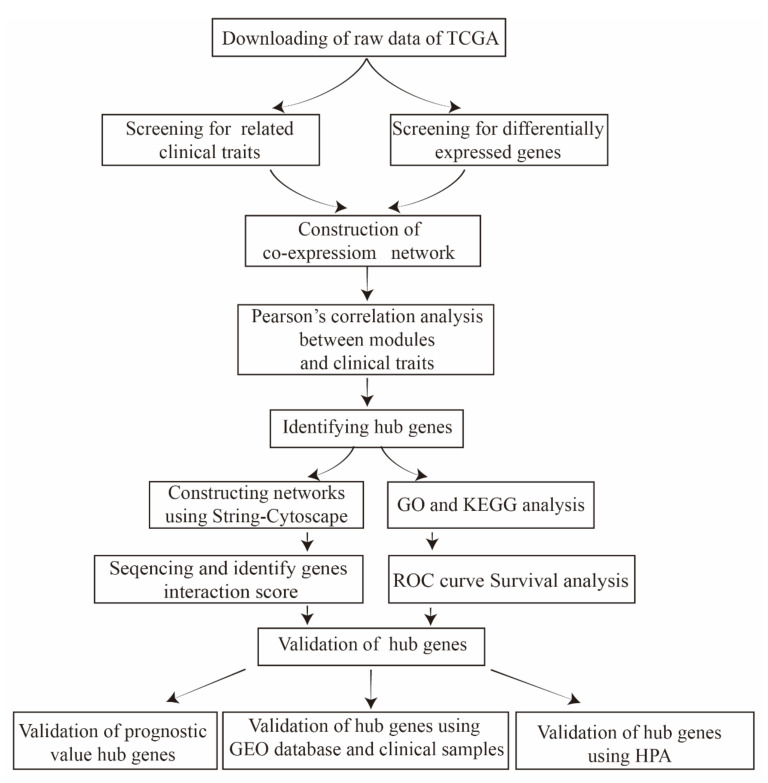
Flowchart of data collection and processing.

**Figure 2 biomedicines-11-00847-f002:**
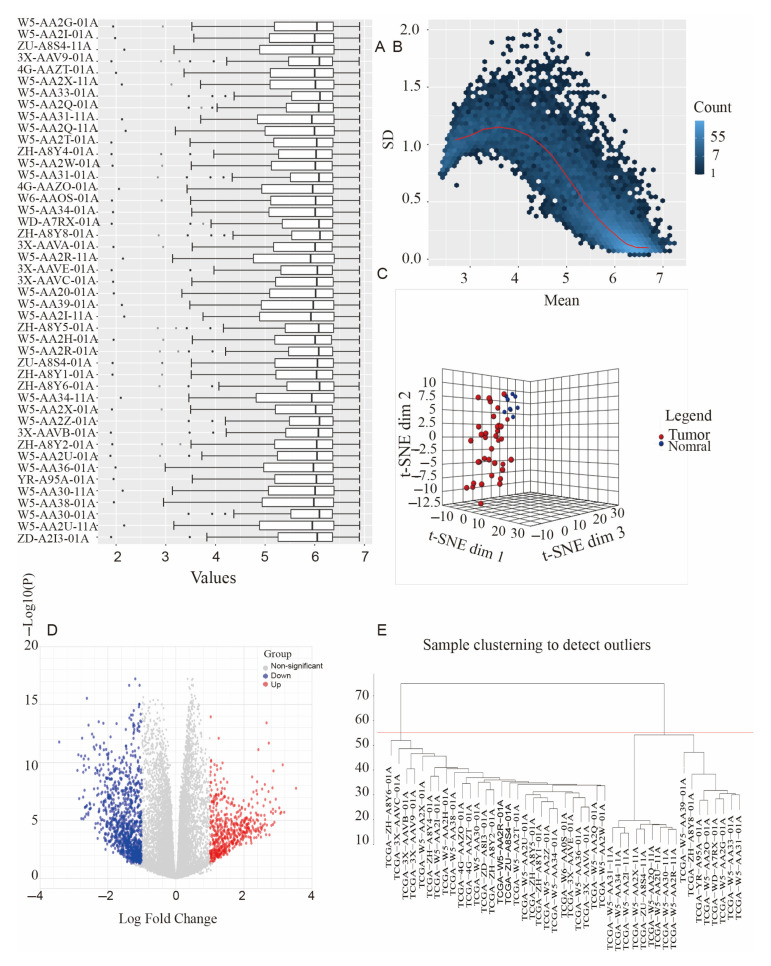
Identification of DEGs for analysis. (**A**) Box plot of 45 samples after normalization (log2-counts per million, variance filter: 15; low abundance: 4). (**B**) MSD plot of 45 samples after normalization. (**C**) t-Distributed stochastic neighbor embedding of DEGs demonstrated that CCA patients and nontumor tissues would be divided into two groups. (**D**) Volcano plot of 1613 DEGs in between CCA patients and normal tissue (adjusted *p*-value < 0.05; |log2fold change| > 1). (**E**) Plot of sample clustering. Set model parameters “cutHigh” and “minSieze” were set at 55 and 10, which ultimately yielded 28 samples.

**Figure 3 biomedicines-11-00847-f003:**
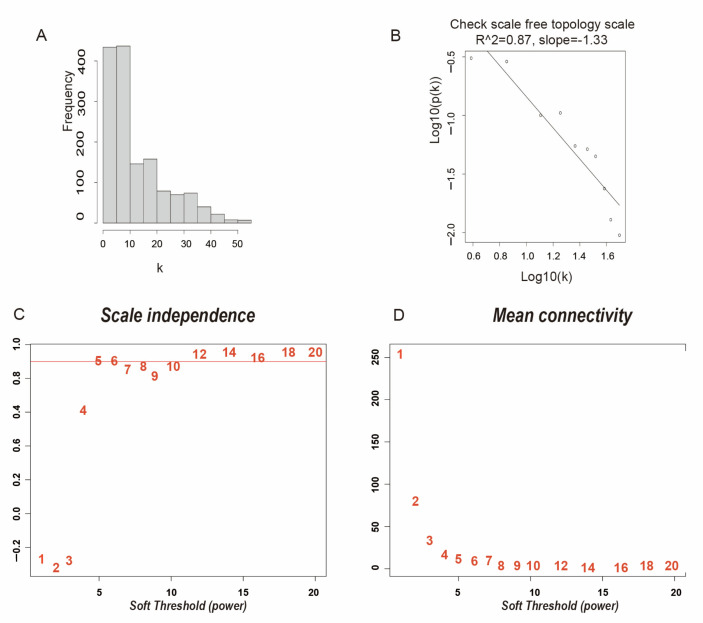
Weighting coefficient β selection. (**A**) Plot of scale-free fit index. (**B**) Plot of different soft-thresholding powers with different mean connectivity. (**C**) Connectivity distribution of nodes with β set as 5. (**D**) The verification of scale-free topology with β set as 5.

**Figure 4 biomedicines-11-00847-f004:**
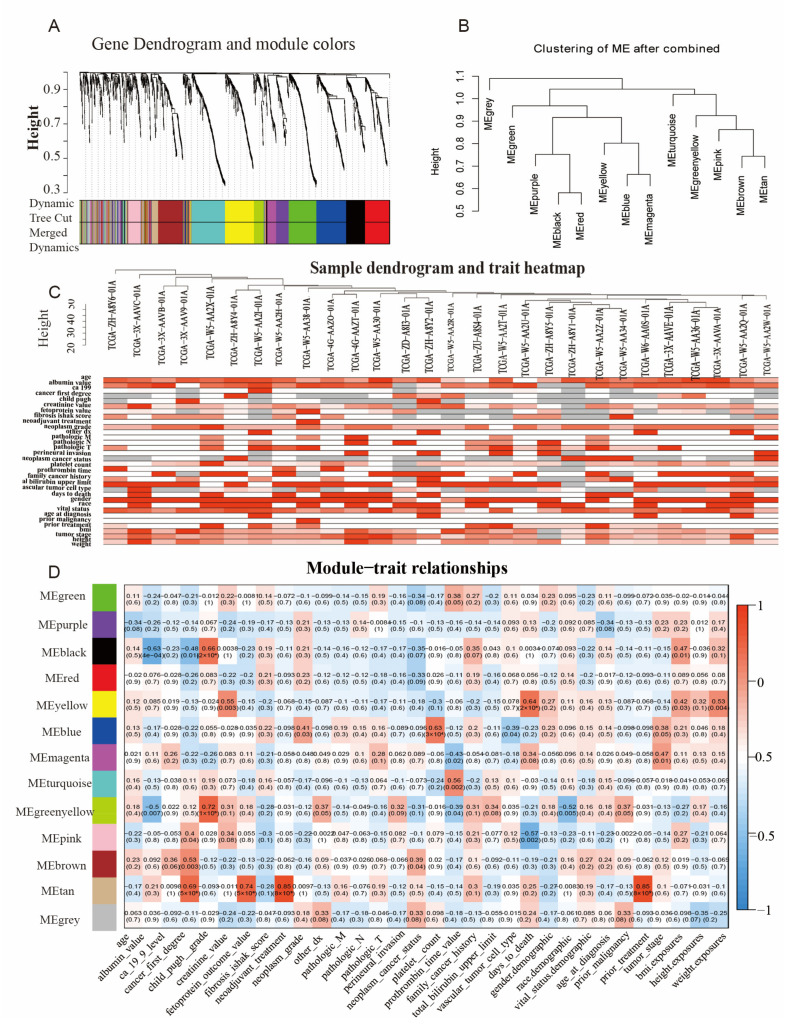
Construction of WGCNA network. (**A**) Dendrogram of gene topological matrix branch (MEs with similarity > 0.75 were merged). (**B**) Dendrogram and heatmap of 13 eigengenes. Dynamic tree of the merged MEs with similarity greater than 0.75. (**C**) Clinical modules were clustered, and correlations were investigated with a sample dendrogram. The color of the heat indicated the correlation value. With respect to clinical data, red represents a high value, white means a low value, and gray means a missing value. (**D**) Heatmap of correlations between MEs and clinical features of CCA. The correlation coefficient (cor) and corresponding *p*-value (*p*-value) between different gene modules and clinical traits. According to the size of correlation coefficient (cor), the darker red represents positive correlation, and the darker blue represents negative correlation.

**Figure 5 biomedicines-11-00847-f005:**
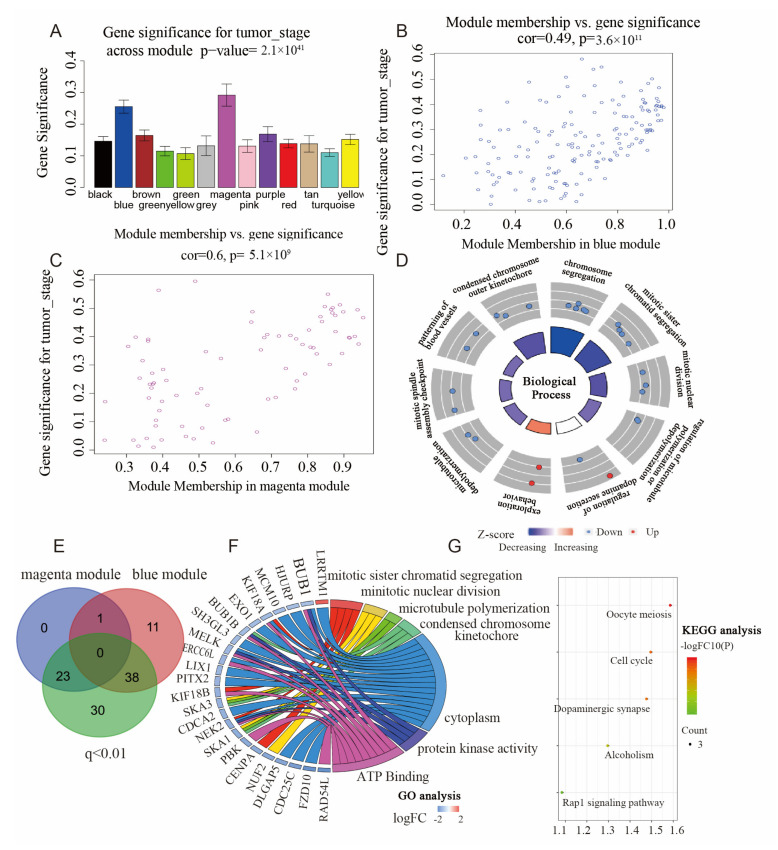
Functional annotation of hub genes. (**A**) GS for tumor stage across each module. (**B**) The plot of GS for tumor stage with MM in blue modules. (**C**) The plot of GS for tumor stage with MM in magenta modules. (**D**) In the genes of blue and magenta modules represented (Appendix A), bubble plot of top 10 biological process by GO enrichment analysis. (**E**) Venn plot of 61 screened hub genes (blue and magenta modules: GS > 0.2, MM > 0.8, and q < 0.01). (**F**) In 61 screened hub genes, GO chord plot demonstrated the top 7 GO enrichment analysis and 24 featured genes. (**G**) In 61 screened hub genes, bubble diagram of pathway enrichment analysis.

**Figure 6 biomedicines-11-00847-f006:**
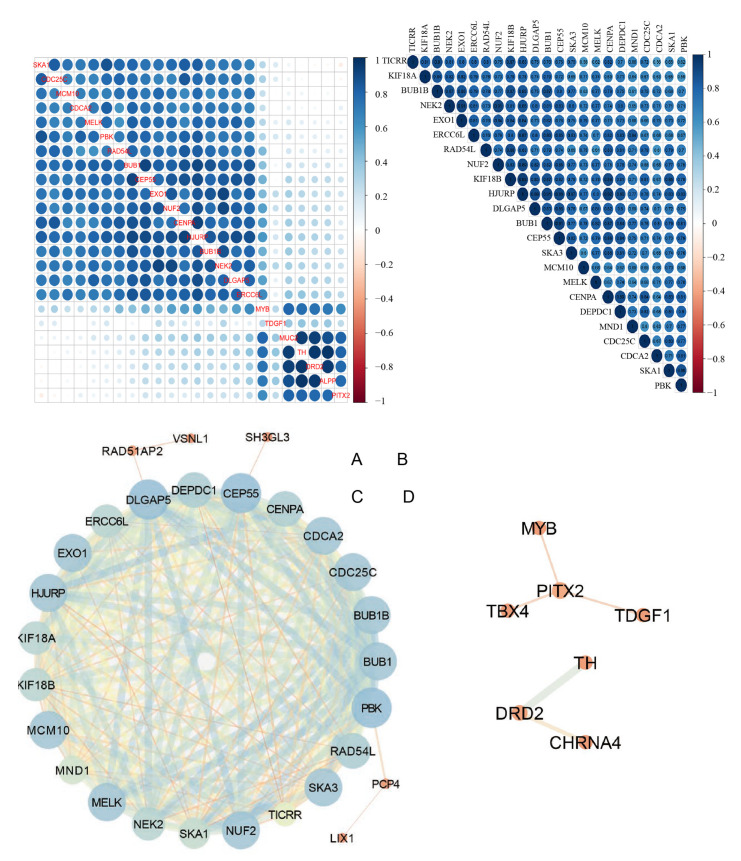
Hub gene analysis. (**A**) Subgroup of co-expression heatmap of 33 hub genes with coefficients annotated. (**B**) Co-expression heatmap of 24 genes with cell cycle. (**C**,**D**) Construction PPI network of co-expression network of 61 hub genes in CCA patients.

**Figure 7 biomedicines-11-00847-f007:**
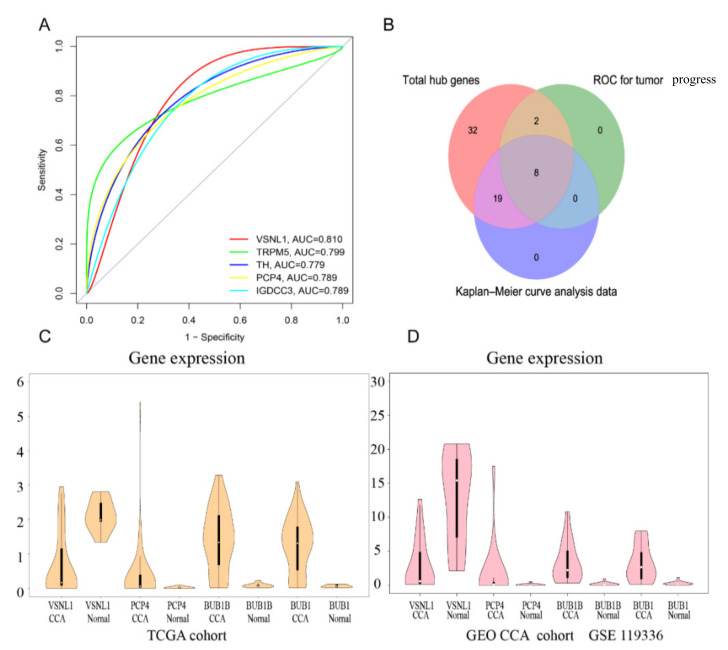
Analysis of hub genes. (**A**) ROC curve analysis showed that five hub genes could distinguish tumorigenic progression of CCA. (**B**) Venn plot of eight hub genes. (**C**) mRNA expression of 4 hub genes of 36 patients with CCA, compared to the 9 adjacent tissues, according to the TCGA CCA database. (**D**) mRNA expression of 4 hub genes in 15 CCA patients and 15 adjacent tissues, according to the GSE119336 cohort.

**Figure 8 biomedicines-11-00847-f008:**
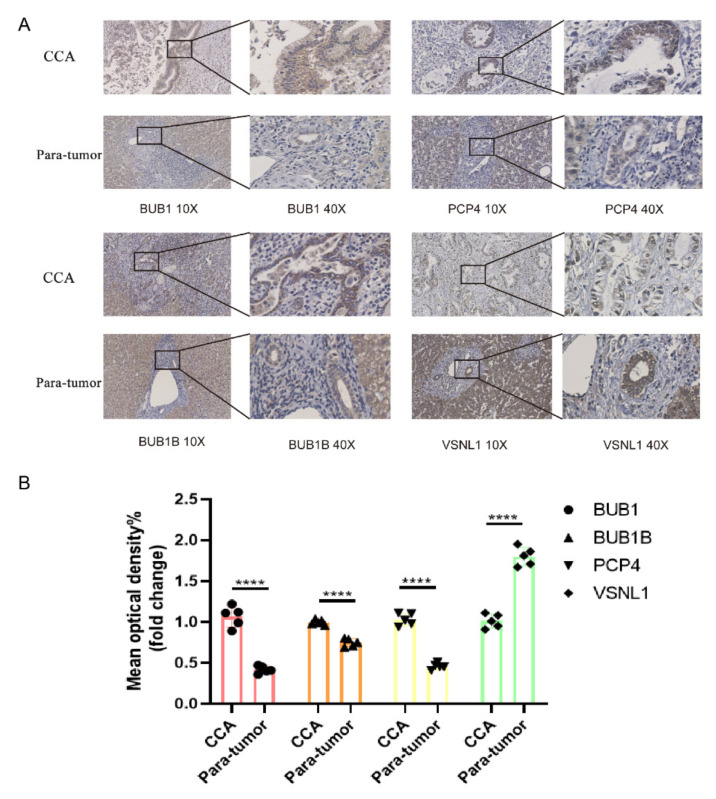
Verification of hub genes. (**A**) Immunohistochemical staining of CCA and adjacent tissue samples. Blue indicates nuclear staining, and brown indicates target protein staining. The darker the brown, the more the target protein was expressed. The expression of BUB1, BUB1B, and PCP4 were significantly increased in CCA patient tissue, but the protein level of VSNL1 was significantly lower in CCA tissue. (**B**) Mean optical density of the expression of BUB1, BUB1B, PCP4, and VSNIL1 of cholangiocarcinoma cells in CCA and bile duct cells in adjacent tissue samples. Staining areas within normal hepatocytes were excluded. **** *p* < 0.0001.

## Data Availability

The TCGA cohort data and GSE119336 cohort data are available from UCSC Xena (https://xenabrowser.net), accessed on 26 February 2021, and the Gene Expression Omnibus public repository. The original contributions presented in the study are included in the article.

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
