# Peer review of "Characteristic Analysis of Featured Genes Associated with Cholangiocarcinoma Progression"

_biomedicines, 2023, doi:10.3390/biomedicines11030847_

Round 1

Reviewer 1 Report

This work aimed to identify genes that are associated with the progression of CCA in the hope to find some new biological markers for CCA diagnosis. Using the TCGA dataset, the authors constructed 13 co-expression modules through WGCNA analysis and then focused on the modules that were significantly associated with tumor stage. The authors further identified 8 hub genes out of 61 featured genes that are highly correlated with CCA progression and prognosis. Here are some points that need to be addressed.

1. Language editing is necessary to correct all the typos and grammar mistakes. For example, “CAA” should be replaced with “CCA”; 2 explanations for Figure 7A were found in the legend.

2. Why is MEgrey not included in Figure 4B?

3. The plots for Figure 4C and Figure 4D were placed in the wrong order.

4. I would suggest the authors explain why VSNL1, PCP4, BUB1 and BUB1B were chosen for experimental validation since they have identified a total of 8 hub genes that are closely related to tumor progression and prognosis.

5. I would suggest the authors make some comparison between their own findings with other researches in regard to the 8 hub genes identified in Figure 7B, especially the 4 genes selected for experimental validation.

5. The authors have made some basic introduction for the 4 genes selected for experimental validation in the discussion section. I would suggest the authors make some further discussion about the implication of the potential roles of these genes in CCA progression.

Reviewer 2 Report

Title: Characteristic Analysis of Featured Genes Associated Cholangiocarcinoma Progression

This paper describes 61 hub genes that regulate 26 the progression and prognosis of CCA. Eight hub genes (VSNL1, TH, PCP4, IGDCC3, RAD51AP2, 27 MUC2, BUB1, and BUB1B) were identified, which exhibited significant interactions with the tumor-28 igenic mechanism and prognosis of CCA. This paper includes a relatively large amount of data, so the results may be highly reliable.

The manuscript cites many references and is well written. However, there are some questions and the author is requested to add the descriptions according to comments as below.

1. Verification of hub genes  

Regarding to the experimental results of immunohistochemistry, how did you assess the the protein level of VSNL1? Authors need to indicate the details of methods. 

2. Venn plot of 10 hub genes

Regarding to venn plot of 10 hub genes, what`s the definition of ROC for tumor progression as well as Kaplan-Meier curve analysis data? 

Reviewer 3 Report

In this manuscript, Yao et al. demonstrated the utilization of bioinformatics tools to identify Cholangiocarcinomas (CCA) diagnostic markers, thereby aiding in innovating a non-invasive diagnostic approach for CCA.

The manuscript is interesting. The bioinformatics pipeline is logical. However, I have the following concerns and suggestions before considering this manuscript for publication:

1- Extensive language editing is required.

2- The purpose of the GO analysis in Figure 3A-C is unclear. It generates random hits that are not useful for the remainder of the manuscript (e.g., neurotransmitter binding). I recommend removing this data from the manuscript.

3- It needs to be clarified what the difference is between the dendrogram in Figure 4B and Figure S3C. 

4- What is the difference between the two dendrograms in Figure S3C?

5- The gene modules are named based on colors. However, I recommend adding the gene names in each module (color) as a supplementary table which will serve as an excellent database for CCA researchers.

6- The heat map in Figure 4C needs to be clarified. The numbers need to be explained in the figure legend. What is the color of the heat indicating? P-value or correlation value? You need to explain more in detail. Moreover, no need to show all the numbers in every square of the heatmap. Highlight the significant ones only. All values can be given as a supplementary table for interested readers.

7- Figure 4D needs to be clarified. It is not easy to read the words, and it overlaps the x-axis of the heat map in Figure 4C. There is no indication of what is correlated between the patients and the y-axis of the heat map.

8- Figure 6 C-D, the networks do not display any variation in width or color. All the nodes and edges look the same and don't match what is indicated in the main text.

9- Figure S4 is not at all clear to read. I recommend highlighting a few critical genes. Values could be added as a supplementary table for all the correlations.

10- A highly recommended analysis is to score these genes across CCA libraries in the single-cell atlas or single-cell portals. This analysis can aid in demonstrating whether the gene modules score high in the malignant cells of the tumor environment in comparison to stromal and immune cells. Such findings will significantly increase the impact of the paper and demonstrate the robustness of the techniques and called hits.

Round 2

Reviewer 2 Report

Title: Characteristic Analysis of Featured Genes Associated Cholangiocarcinoma Progression

This paper describes 61 hub genes that regulate 26 the progression and prognosis of CCA. Eight hub genes (VSNL1, TH, PCP4, IGDCC3, RAD51AP2, 27 MUC2, BUB1, and BUB1B) were identified, which exhibited significant interactions with the tumorigenic mechanism and prognosis of CCA. This paper includes a relatively large amount of data, so the results may be highly reliable.

The manuscript revised is well written. I believe this manuscript is acceptable.

Reviewer 3 Report

The authors have indeed addressed many of my comments. The manuscript in its current format is improved significantly. It is undoubtedly significant to associate novel genes with CCA. Using computational tools, the authors have demonstrated a method to obtain such potential markers. I have the following minor comments which should be addressed before considering this manuscript for publication:

1- The English language and style in the manuscript are improved, but there are still a few grammatical errors. Please make sure to correct them. (Examples in the abstract itself)

2- In the bubble plot (Figure 5D), replacing the GO IDs with the abbreviation of the terms will be better.

3- For the Figure 5D legend, please indicate which genes were subjected to this analysis in the legend itself. There are two GO analyses in the same figure, so it will be nice to distinguish them from the figure legend itself rather than going back and forth between the legend and the main text.
